# Identification of Androgen Receptor Metabolic Correlome Reveals the Repression of Ceramide Kinase by Androgens

**DOI:** 10.3390/cancers13174307

**Published:** 2021-08-26

**Authors:** Laura Camacho, Amaia Zabala-Letona, Ana R. Cortazar, Ianire Astobiza, Asier Dominguez-Herrera, Amaia Ercilla, Jana Crespo, Cristina Viera, Sonia Fernández-Ruiz, Ainara Martinez-Gonzalez, Veronica Torrano, Natalia Martín-Martín, Antonio Gomez-Muñoz, Arkaitz Carracedo

**Affiliations:** 1Center for Cooperative Research in Biosciences (CIC bioGUNE), Basque Research and Technology Alliance (BRTA), Bizkaia Technology Park, Building 801A, 48160 Derio, Spain; lcamacho@cicbiogune.es (L.C.); azabala@cicbiogune.es (A.Z.-L.); acortazar@cicbiogune.es (A.R.C.); iastobiza@cicbiogune.es (I.A.); aercilla.ciberonc@cicbiogune.es (A.E.); jcrespo@cicbiogune.es (J.C.); cviera@cicbiogune.es (C.V.); sfernandez@cicbiogune.es (S.F.-R.); amgonzalez@cicbiogune.es (A.M.-G.); vtorrano@cicbiogune.es (V.T.); nmartin@cicbiogune.es (N.M.-M.); 2Biochemistry and Molecular Biology Department, University of the Basque Country, 48040 Bilbao, Spain; asidomiherre@gmail.com (A.D.-H.); antonio.gomez@ehu.es (A.G.-M.); 3Centro de Investigación Biomédica En Red de Cáncer (CIBERONC), 28029 Madrid, Spain; 4IKERBASQUE, Basque Foundation for Science, 48009 Bilbao, Spain

**Keywords:** prostate cancer, bioinformatics, mouse models, sphingolipid metabolism, ceramide kinase

## Abstract

**Simple Summary:**

Prostate cancer cells require androgens to survive and grow. In turn, targeting androgen signaling has become a predominant therapeutic strategy in this disease. These hormones regulate a plethora of biological processes, which identification could aid the refinement of future anticancer treatments. Our aim was to uncover metabolic processes under the control of androgens, taking advantage of bioinformatics analyses using publicly accessible data in prostate cancer. We found that these hormones control the abundance of an enzyme, ceramide kinase (CERK). CERK produces ceramide-1-phosphate, a metabolite with prosurvival and migration properties. This finding suggests that antiandrogen therapies could be limited by the reactivation of this metabolic process.

**Abstract:**

Prostate cancer (PCa) is one of the most prevalent cancers in men. Androgen receptor signaling plays a major role in this disease, and androgen deprivation therapy is a common therapeutic strategy in recurrent disease. Sphingolipid metabolism plays a central role in cell death, survival, and therapy resistance in cancer. Ceramide kinase (CERK) catalyzes the phosphorylation of ceramide to ceramide 1-phosphate, which regulates various cellular functions including cell growth and migration. Here we show that activated androgen receptor (AR) is a repressor of *CERK* expression. We undertook a bioinformatics strategy using PCa transcriptomics datasets to ascertain the metabolic alterations associated with AR activity. *CERK* was among the most prominent negatively correlated genes in our analysis. Interestingly, we demonstrated through various experimental approaches that activated AR reduces the mRNA expression of *CERK*: (i) expression of *CERK* is predominant in cell lines with low or negative AR activity; (ii) AR agonist and antagonist repress and induce *CERK* mRNA expression, respectively; (iii) orchiectomy in wildtype mice or mice with PCa (harboring prostate-specific *Pten* deletion) results in elevated *Cerk* mRNA levels in prostate tissue. Mechanistically, we found that AR represses *CERK* through interaction with its regulatory elements and that the transcriptional repressor EZH2 contributes to this process. In summary, we identify a repressive mode of AR that influences the expression of *CERK* in PCa.

## 1. Introduction

Hormone signaling governs the molecular activity of an important fraction of cells in our body. Steroid hormones are essential for the development and function of a wide range of tissues through the regulation of nuclear receptors, including sexual organ development and function [1,2,3]. In male reproductive organs, androgens activate androgen receptors (AR) to elicit a broad transcriptional program [4]. These steroids induce a conformational change in AR that promotes its dimerization, translocation to the nucleus, and association with Androgen Response Elements (ARE) in the DNA [4].

Similar to their normal counterparts, prostate cancer cells require androgen signaling to survive and proliferate [5,6,7]. In turn, androgen receptor and androgen synthesis represent pivotal therapeutic targets (androgen deprivation therapy or ADT), and a variety of anticancer agents targeting this hormonal program has proven to hamper prostate cancer progression [7,8,9]. However, resistance to androgen deprivation often emerges, leading to a form of the disease (castration-resistant prostate cancer or CRPC) that accounts for a large fraction of prostate cancer mortality [10,11,12]. It is therefore essential to deconstruct the molecular events that account for ADT efficacy and drug resistance.

Due to its relevance in normal and cancer cell homeostasis, the study of AR signaling has become a focus of research. Many AR targets have been identified and validated, a process that has advanced exponentially with the implementation of high throughput genomics and transcriptomics technologies [13,14,15]. Whereas the vast majority of AR target genes are transcriptionally activated by androgens, reports point at AR-mediated gene repression as an emerging phenomenon [16,17,18].

Metabolic pathways have been consistently reported among the molecular programs regulated by androgens. Androgens control the uptake and biosynthesis of different types of lipids and the metabolism of amino acids and carbohydrates [19,20,21,22,23,24]. However, a systematic analysis of metabolic routes under androgen regulation is lacking.

Sphingolipids function as second messengers and structural components essential for cell homeostasis [25,26]. Ceramide is at the core of this metabolic pathway, and its role in cell biology is widely accepted [26]. Whereas this metabolite activates tumor-suppressive processes, its phosphorylation and the production of ceramide-1-phosphate (C1P) induces a switch in its activity, thus promoting cell viability, proliferation, and migration [26]. Ceramide kinase (CERK) catalyzes the production of C1P and, in turn, the regulation of this enzyme is paramount in sphingolipid homeostasis [25]. Yet, our understanding of the regulation of CERK in health and disease is limited, and upstream regulatory cues need to be identified.

Here we annotate the metabolic genes that are consistently correlated with androgen receptor activity by means of a bioinformatics approach encompassing a compendium of publicly available PCa transcriptomics datasets, which we term the androgen receptor metabolic correlome. Among the regulated genes and pathways, we focus on sphingolipid metabolism and demonstrate that CERK is an AR-repressed gene. We shed light on the molecular mechanism of action of AR to repress CERK, thus expanding the information regarding the mechanism and function of androgens.

## 2. Results

To identify metabolic processes regulated by AR signaling in PCa, we undertook a bioinformatics approach (Figure 1A). First, we set out to define the gene signature that would best illustrate the activation status of AR in prostate cancer. We started from a published meta-analysis of AR signaling-associated gene expression studies, in which the authors identified a gene set that was consistently regulated by androgens in 6 different experimental studies [15]. The positive or negative transcriptional activation by AR was ascertained in that gene set with 34 genes (Appendix A). Stemming from the work by Massie et al. [27], together with Cancertool [28], we could annotate these genes as induced or repressed by activated AR (Appendix A). We identified 23 genes consistently activated by androgens and 8 that were repressed, whereas 3 genes were excluded from the analysis due to the inability to assign directionality. Of note, this shortlist contained well-known AR target genes, including kallikreins KLK2 and KLK3 [29], thus validating our bioinformatics strategy. 

We built an AR activity gene expression signature based on the ratio of expression of upregulated and downregulated AR target genes, which we termed AR signature (AR Signature = (average expression of upregulated genes in Log_2_) − (average expression of upregulated genes in Log_2_); Figure 1A). We next selected the genes identified as metabolic enzymes and transporters according to KEGG [30] and metabolic co-regulators [31,32], leading to a gene set of 2775 genes that we defined as metabolic genes. We performed correlation studies with the metabolic gene set and AR signature in 8 different PCa transcriptomics datasets containing primary tumor specimens. We selected those that exhibited a consistent correlation, meaning a direct (coefficient greater than 0.2 and *p*-value lower than 0.05) or inverse (coefficient lower than −0.2 and *p*-value lower than 0.05) correlation in more than 50% of datasets with available data for a given gene [33,34,35,36,37,38,39,40]. This correlation analysis led to a list of 223 metabolic genes consistently correlated with AR activity, that we defined as the AR metabolic correlome (Figure 1A; Appendix A). As a validation of our strategy, we could confirm that the top metabolic genes directly correlated with AR activity, GPT2 and SLC45A3, are reported AR targets [41,42].

To ascertain the metabolic impact of androgen signaling, we performed gene enrichment analyses. KEGG analysis revealed a series of metabolic pathways enriched in the list of AR correlated genes (Figure 1B). We interrogated the nature and predicted impact of androgen signaling on the resulting metabolic pathways. Some of these pathways have been previously reported as androgen-regulated [19,20,21,22,23,24]. Interestingly, sphingolipid metabolism exhibited a gene expression alteration by androgens that directed the pathway towards the production of ceramide (Figure 1C). AR activity is directly correlated with the expression of enzymes that metabolize sphingolipids to produce ceramide, whereas it is inversely correlated with ceramide kinase, an enzyme that will reduce the pool of ceramide by eliciting its phosphorylation [43]. We decided to focus on this metabolic pathway for subsequent studies, owing to the little information regarding the regulation of the pathway by androgens. A summary of the correlation results of the AR signature with the mRNA expression of these enzymes in multiple PCa datasets is depicted in Figure 1D.

Correlation studies based on activity signatures represent an invaluable tool to identify genes which expression is associated with a pathway of interest. However, this information is insufficient to conclude that such genes are targets of the process under investigation as they could be altered as a secondary consequence of downstream molecular effectors. To ascertain the genes within sphingolipid metabolism that are regulated by androgens, we performed a variety of in vitro and in vivo experimental approaches. We evaluated the expression pattern of these candidate genes in a panel of prostate cell lines with known androgen signaling status. We took advantage of *KLK3* (the gene encoding for prostate-specific antigen, PSA) as a readout of AR activity. As reported, LNCaP, C4-2, 22RV1, and VCaP cells exhibited robust AR activity by means of *KLK3* expression, whereas PC3 and DU145 (AR-negative prostate cancer cells) or RWPE1, PWR1E, and BPH1 cells (benign prostate immortalized cells) exhibited low activity of the nuclear receptor (Figure 2A). Among the five sphingolipid metabolism genes identified in the correlome, only *CERK* and *PLPP1* exhibited a consistent association with AR activity. *PLPP1* exhibited a positive association with androgen signaling, and *CERK* expression was reduced in AR-expressing cells (Figure 2A and Appendix A). Next, we manipulated the activation status of AR in vitro with selective agonists (dihydrotestosterone, DHT) or antagonists (MDV3100, Enzalutamide) [44]. The effect of these compounds on the expression of *KLK3* was robust 24 hours after treatment (Figure 2B). Among the five sphingolipid metabolism genes, *PLPP1* and *CERK* exhibited a significant time-dependent regulation of gene expression after 24 h with both agents, with the predicted directional consistency (Figure 2B and Appendix A).

Androgens control both prostate development and growth [4,5,6]. To corroborate the regulation of sphingolipid metabolic genes in a more complex experimental setting, we performed orchiectomy in adult male mice, in a context of normal prostate physiology (wildtype mice) or PCa (prostate-conditional *Pten* knockout mice). As mice do not express kallikreins, we used the androgen-dependent gene *Nkx3.1* to validate the loss of AR activity after castration (Figure 2C). Among the five genes, only *Cerk* exhibited the predicted changes in mRNA levels upon castration (Figure 2C and Appendix A). We focused our attention on this enzyme and showed that *CERK* was repressed by androgen signaling in a validation set of PCa cell lines (C4-2, VCaP, and 22Rv1) (Appendix A). Of note, CERK was less responsive to androgen regulation in 22RV1, a cell line that expresses an AR variant, and was not regulated in the AR-negative cell line PC3 (Appendix A). These results are in line with the inverse correlation between the AR target *KLK3* and *CERK* in various PCa transcriptomics datasets (Figure 2D and Appendix A). Overall, our results reveal *CERK* as a gene repressed by androgen signaling in PCa.

The regulation of *CERK* expression by androgens should influence the production of C1P. To evaluate whether AR signaling regulates CERK activity and C1P production, we measured the abundance of these metabolites in a context of AR inhibition in androgen-dependent LNCaP cells that exhibit low *CERK* expression. In agreement with the upregulation of CERK observed upon AR inhibition, we detected an increase in C20 and C24 C1P levels, but not in their non-phosphorylated ceramide counterparts (Figure 2E and Appendix A). Of note, the levels of sphingosine-1-phosphate, a different sphingolipid not phosphorylated by CERK, remained unaffected by MDV3100 (Appendix A).

To elucidate the mechanism of repression of CERK, we focused on the regulatory activity of AR. This nuclear receptor functions predominantly as a transcriptional activator. However, there are examples of genes repressed by androgen signaling [16,17,45,46]. We performed chromatin immunoprecipitation (ChIP) assays. We studied two potential androgen receptor binding sites (AR sites) in the proximity of the *CERK* gene based on a previous report (Figure 3A) [47]. Only AR site 1 exhibited significant binding to AR, and androgen stimulation significantly increased the association of the nuclear receptor to this site (Figure 3B,C). As a quality control of our assay, we gathered two results. First, we showed that androgens enhanced the binding of AR to the AR sites present in *KLK3* by ChIP (Figure 3C). Second, AR binding to AR site 2 did not show enrichment in the qPCR analysis, suggesting that the assay did not have technical issues that would artificially provide binding to *CERK* (Figure 3B).

The repressive transcriptional activity of AR has been linked to the function of the chromatin remodeler EZH2 [45], among other factors. We hypothesized that EZH2, the functional enzymatic component of the Polycomb Repressive Complex 2, could function as a CERK repressor and that this protein would define the negative regulation of the sphingolipid metabolic gene by androgens. On the one hand, we showed that EZH2 was significantly associated with the AR-binding region in *CERK* AR site 1 (Figure 3D). On the other hand, we took advantage of a well-characterized inhibitor of EZH2, GSK126 [48]. Treatment with this compound in PCa cells revealed a dose-dependent de-repression of *CERK* (Figure 3E). Moreover, the EZH2 inhibitor partially prevented DHT-induced *CERK* repression, whereas it cooperated with AR antagonist MDV3100 in the activation of CERK expression (Figure 3E). Of note, this inhibitor did not exhibit a consistent effect on the regulation of the AR canonical target *KLK3* (Appendix A), in line with the non-repressive action of AR on this kallikrein.

EZH2 expression is positively associated with PCa pathogenesis and progression [49]. We interrogated whether the regulation of *CERK* by *EZH2* could be a general phenomenon in this tumor type. To this end, we studied the correlation between *CERK* and *EZH2* mRNA abundance. Interestingly, we observed an inverse correlation between the expression of the two genes in various PCa cohorts (Figure 3F and Appendix A). We then evaluated whether this regulation could influence the expression of *CERK* in PCa. Analysis of available PCa datasets confirmed that *EZH2* was predominantly upregulated in localized PCa compared to normal prostate (Figure 3G), and this phenomenon was associated with a robust downregulation of *CERK* in the same datasets. These results strongly suggest that AR represses *CERK*, at least in part, through EZH2, and that the epigenetic modifier might be a general regulator of *CERK* expression beyond AR.

We and others have previously shown that the product of CERK, C1P, promotes cancer cell aggressiveness in other cellular systems [50]. Therefore, we tested the influence of C1P in prostate cancer cell function. First, we generated spheroids of LNCaP cells. These spheroids were embedded in collagen, challenged with vehicle or C1P (20 µM), and the increase in the total area of the structure was evaluated 7 days later. Interestingly, C1P treatment did not influence two-dimensional cell growth but increased spheroid diameter (Figure 4A,B). Second, C1P treatment (20 µM) slightly increased the migration of LNCaP cells in wound healing assays (Figure 4C). Third, we performed transwell migration assays in a migratory prostate cancer cell line, namely PC3. C1P elicited a time-dependent increase in cell migration, over a monitored period of 72 h (Figure 4D). As a control of this assay, we tested the effect of the non-phosphorylated counterpart (C_2_-Ceramide, C2C) on PC3 cells. As reported [50] we corroborated the growth-suppressive activity of this sphingolipid (Appendix A).

To further ascertain the role of CERK in PCa cell biology we set up genetic manipulation tools to perturb *CERK* expression. We chose to silence *CERK* in PCa cells with high and low expression of this gene, PC3 and LNCaP, respectively. We generated two independent inducible lentiviral shRNA sequences that efficiently reduced *CERK* mRNA abundance (Appendix A). In agreement with our C1P supplementation results, *CERK* silencing reduced aggressive properties without affecting cell proliferation in PC3 cells, but had no functional consequences in *CERK* low expressing cells, LNCaP (Appendix A).

These results reinforce the notion that ceramide phosphorylation by CERK could have a biological impact on prostate cancer cells.

## 3. Discussion

The signaling cascade regulated by AR is essential for the function of prostate cells [6,7,9]. The studies of AR transcriptional regulation were limited to a few AR-target genes but thanks to the advances in high-throughput genomic technologies, now we know that this nuclear receptor regulates the expression of more than 1500 genes [15,47]. Androgen-bound AR is primarily known as a transcription activator but additional evidence shows that it can also repress gene expression [16,17]. Only a handful of genes have been molecularly characterized as directly repressed by AR [17,45,51]. In this context, our results provide detailed molecular regulation of a gene repressed by AR, with validations that encompass human specimens, murine models, and cellular systems. In turn, our work, together with previous studies by other groups, encourages a more extensive analysis of the molecular determinants of AR-elicited transcriptional activation vs. repression.

Lipid uptake and metabolism is under the control of androgen signaling [20,52,53,54,55,56,57]. In fact, elevated lipid biosynthesis has been postulated to promote the development of castration-resistant PCa [52,54], and inhibition of fatty acid synthase has been proposed as a therapeutic strategy to target this pathological state [53]. Despite the vast knowledge around lipid metabolism regulation by androgens, the information around the role and relevance of these hormones in the control of sphingolipid metabolism is limited. Sphingolipid metabolism produces metabolites that influence multiple biological processes [26,58,59]. Phosphorylation of ceramide species illustrates the dichotomy of sphingolipids in the regulation of cell survival and motility vis a vis cell death [50,60]. In this regard, CERK activity is predicted to support cell viability, with an increase in the pool of phosphorylated ceramide. Our results corroborate that C1P can promote prostate cancer cell aggressiveness. Since AR represses CERK, the functional outcome of this regulation would be a decrease in C1P. Conversely, our results suggest that loss of AR signaling derepresses CERK and leads to the elevation of C1P, which could counteract the therapeutic effect of AR antagonists and uncover a mechanism for the development of castration-resistant PCa. Moreover, this increase in CERK levels could represent an unprecedented target to curb prostate cancer aggressiveness, as demonstrated in AR-negative PC3 cells. The specific pathophysiological context in which this regulation be therapeutically exploited remains to be defined.

EZH2 is an epigenetic modifier that is overexpressed in multiple cancer types [61]. In PCa, this gene is associated with disease progression and metastasis [49], and the crosstalk between AR and this factor has been previously reported [45,62,63,64]. Interestingly, we report that EZH2 represses *CERK*. EZH2 inhibitor GSK126 elicited robust activation of *CERK* expression, and combination with AR agonists and antagonists interfered with the repressive activity of androgen signaling on this gene. Moreover, EZH2 interacted with the androgen receptor site in *CERK* regulatory region. Our analysis of *CERK* and *EZH2* gene expression in PCa datasets suggests that the action of the epigenetic modifier on *CERK* might be operational beyond the context of AR. In turn, our results open a new area for the study of Polycomb group proteins in the control of sphingolipid metabolism.

## 4. Materials and Methods

### 4.1. Cell Culture

Human prostate cancer cell lines PC3, DU145, LNCaP and the benign prostate hyperplasia cell line BPH1 were purchased from Leibniz Institut DSMZ (Deutsche Sammlung von Mikroorganismen und Zellkulturen GmbH), who provided authentication certificate. Human prostate cancer cell lines 22RV1 and VCaP were purchased from American Type Culture Collection (ATCC). Human prostate cancer cell line C4-2, and human prostatic epithelial cell lines PWR1E and RWPE1 were generously provided by the laboratory of Dr. Pier Paolo Pandolfi. Cell lines were periodically subjected to microsatellite-based identity validation. The cell lines used in this study were not found in the database of commonly misidentified cell lines maintained by ICLAC and NCBI Biosample. All cell lines were routinely monitored for mycoplasma contamination. DU145, PC3, and VCaP cell lines were maintained in Dulbecco’s Modified Eagle Medium (DMEM) while LNCaP, C4-2, and 22RV1 cell lines were maintained in regular Roswell Park Memorial Institute (RPMI) medium without any supplement except for complete 10% (*v*/*v*) fetal bovine serum (FBS, Gibco) and 1% (*v*/*v*) penicillin-streptomycin (Gibco). PWR1E and RWPE1 cell lines were maintained in Keratinocyte Serum-Free Medium (K-SFM; Gibco) supplemented with 0.05 mg/mL bovine pituitary extract (BPE; Gibco) and 5 ng/mL epidermal growth factor (EGF; Gibco). BPH1 was maintained in RPMI medium supplemented with 20% FBS, 20 ng/mL DHT, 5 ug/mL transferrin, 5 µg/mL insulin, and 5 ng/mL sodium selenite (all supplements from Sigma-Aldrich St. Louis, MO, USA).

### 4.2. Generation of Stable Cell Lines

293FT cells were used for lentiviral production. Lentiviral vectors expressing validated shRNAs against human CERK from the Mission shRNA Library were subcloned in a Plko Tet on inducible system (following the strategy provided by Dr. Dmitri Wiederschain [65], Addgene plasmid # 21915). Cells were transfected with lentiviral vectors following standard procedures, and viral supernatant was used to infect cells. The selection was done using puromycin (2 µg/mL) for 48 h. As a control, a lentivirus with scrambled shRNA (shC) was used. (Short hairpins sequences: sh88: CCGGGCCACGATGGATCGCTGGTTTCTCGAGAAACCAGCGATCCATCGTG GCTTTTTG, sh89: CCGGCGGCTTAAACTTTGATCTGTACTCGAGTACAGA TCAAAGTTTAAGC CGTTTTTTG, shC: CCGGCAACAAGATGAAGAGCACCAACTCGAGTTGGTGCTCTTCATCTTG TTG.

### 4.3. Reagents

DHT (4,5α-Dihydrotestosterone) was purchased from Sigma-Aldrich (St. Louis, MO, USA; A8380) and dissolved in ethanol at 10 μM to be used at a final concentration of 10 nM. MDV3100 (enzalutamide), from Santa Cruz Biotechnology (Dallas, TX, USA; sc-364354), was diluted in DMSO at 10 mM and used at 10µM. GSK126 (EZH2 inhibitor) was purchased from MedChemExpress (Monmouth Junction, NJ, USA; HY-13470), dissolved in DMSO, and used in a range of concentrations for 24 h. C2 Ceramide (N-acetoyl-D-erythro-sphingosine) was purchased from Avanti Polar Lipids (Alabaster, AL, US; 860502), dissolved in DMSO, and used at a final concentration of 20 μM. Doxycycline Hyclate (Dox, used at a final concentration of 0.25 μg/mL) and C16 Ceramide 1-phosphate (C1P, 860533P) were purchased from Sigma Aldrich (St. Louis, MO, US). An aqueous dispersion (in the form of liposomes) was prepared by sonicating C1P (1 mg) in sterile nanopure water (600 μL) on ice using a probe sonicator for six cycles of 8 s on and 5 s off until a clear dispersion was obtained. C1P concentration in the stock solution was 2.6 mM and a final concentration of 20 μM was used for all cellular assays. This procedure is considered preferable to C1P dispersions in organic solvents because lipid droplet formation is minimized and exposure of cells to alcohols or dodecane is avoided.

### 4.4. Animals

Prostate epithelium-specific deletion *Pten* knockout (C57/BL6/129sv; Pb-Cre4; *Pten* lox/lox) model was kindly provided by Dr. Pandolfi. All mouse experiments were carried out following the ethical guidelines established by the Biosafety and Welfare Committee at CIC bioGUNE (Spanish acronym for center for cooperative research in Biosciences). The procedures employed were carried out following the recommendations from AAALAC (Association for Assessment and Accreditation of Laboratory Animal Care). Orchiectomy was performed in 4 months *Pten^pc+/+^* mice and 6 months *Pten^pc−/−^* mice, and 6 days later prostate lobules were collected. Mice were fasted for 6 h prior to tissue harvest to prevent metabolic alterations due to immediate food intake.

### 4.5. qRTPCR

RNA was extracted using a NucleoSpin^®^ RNA isolation kit (Macherey-Nagel, Dueren, Germany; 740955.240C). For animal tissues, a Trizol-based implementation of the NucleoSpin^®^ RNA isolation kit protocol (Provided above) was used as referenced [66]. For all cases, 1 μg of total RNA was used for cDNA synthesis using Maxima™ H Minus cDNA Synthesis Master Mix (ThermoFisher, Waltham, MA, US; M1682). Quantitative Real-Time PCR (qRTPCR) was performed as previously described [67]. Universal Probe Library (Roche, Basel, Switzerland) primers and probes employed (Roche; ThermoFisher) are detailed in Appendix A. All qRTPCR data presented was normalized using GAPDH/*Gapdh* (ThermoFisher; Hs02758991_g1, Mm99999915_g1).

### 4.6. Sphingolipid Metabolic Analysis

LNCaP cells were treated with vehicle or MDV for 24 h and two million cells per condition were washed, pelleted and sent for analysis. Internal standards were purchased from Avanti Polar Lipids (Alabaster, AL, USA) and were added to samples in 10 µL ethanol:methanol:water (7:2:1) as a cocktail of 250 pmol each. Standards for sphingoid bases and sphingoid base 1-phosphates were 17-carbon chain length analogs: C17-sphingosine, (2S,3R,4E)-2-aminoheptadec-4-ene-1,3-diol (d17:1-So); C17-sphinganine, (2S,3R)-2-aminoheptadecane-1,3-diol (d17:0-Sa); C17-sphingosine 1-phosphate, heptadecasphing-4-enine-1-phosphate (d17:1-So1P); and C17-sphinganine 1-phosphate, heptadecasphinganine-1-phosphate (d17:0-Sa1P). Standards for N-acyl sphingolipids were C12-fatty acid analogs: C12-Cer, N-(dodecanoyl)-sphing-4-enine (d18:1/C12:0); C12-Cer 1-phosphate, N-(dodecanoyl)-sphing-4-enine-1-phosphate (d18:1/C12:0-Cer1P); C12-sphingomyelin, N-(dodecanoyl)-sphing-4-enine-1-phosphocholine (d18:1/C12:0-SM); and C12-glucosylceramide, N-(dodecanoyl)-1-β-glucosyl-sphing-4-eine.

Sample homogenates were collected into 13 × 100 mm borosilicate tubes with a Teflon-lined cap (catalog #60827-453, VWR, West Chester, PA, USA). Then 2 mL of CH3OH and 1 mL of CHCl3 were added along with the internal standard cocktail (250 pmol of each species dissolved in a final total volume of 10 μL of ethanol:methanol:water 7:2:1). The contents were dispersed using an ultra sonicator at room temperature for 30 s. This single-phase mixture was incubated at 48 °C overnight. After cooling, 150 µL of 1 M KOH in CH3OH was added and, after brief sonication, incubated in a shaking water bath for 2 h at 37 °C to cleave potentially interfering glycerolipids. The extract was brought to neutral pH with 12 µL of glacial acetic acid, then the extract was centrifuged using a table-top centrifuge, and the supernatant was removed by a Pasteur pipette and transferred to a new tube. The extract was reduced to dryness using a Speed Vac. The dried residue was reconstituted in 0.5 mL of the starting mobile phase solvent for LC-MS/MS analysis, sonicated for ca 15 s, then centrifuged for 5 min in a tabletop centrifuge before the transfer of the clear supernatant to the autoinjector vial for analysis.

For LC-MS/MS analyses, a Shimadzu Nexera LC-30 AD binary pump system coupled to a SIL-30AC autoinjector and DGU20A5R degasser coupled to an AB Sciex 5500 quadrupole/linear ion trap (QTrap) (SCIEX, Framingham, MA, USA) operating in a triple quadrupole mode was used. Q1 and Q3 were set to pass molecularly distinctive precursor and product ions (or a scan across multiple *m*/*z* in Q1 or Q3), using N2 to collisionally induce dissociations in Q2 (which was offset from Q1 by 30–120 eV); the ion source temperature set to 500 °C.

The compounds were separated by reverse-phase LC using a Supelco 2.1 (i.d.) × 50 mm Ascentis Express C18 column (Sigma, St. Louis, MO, USA) and a binary solvent system at a flow rate of 0.5 mL/min with a column oven set to 35 °C. Prior to injection of the sample, the column was equilibrated for 0.5 min with a solvent mixture of 95% Moble phase A1 (CH3OH/H2O/HCOOH, 58/41/1, *v*/*v*/*v*, with 5 mM ammonium formate) and 5% Mobile phase B1 (CH3OH/HCOOH, 99/1, *v*/*v*, with 5 mM ammonium formate), and after sample injection (typically 40 μL), the A1/B1 ratio was maintained at 95/5 for 2.25 min, followed by a linear gradient to 100% B1 over 1.5 min, which was held at 100% B1 for 5.5 min, followed by a 0.5 min gradient return to 95/5 A1/B1. The column was re-equilibrated with 95:5 A1/B1 for 0.5 min before the next run.

### 4.7. ChIP

Chromatin Immunoprecipitation (ChIP) was performed using the SimpleChIP^®^ Enzymatic Chromatin IP Kit (Cell Signalling Technology, Inc, Danvers, MA, USA; 9003) following the manufacturer’s instructions. Three million LNCaP cells were seeded in 150 mm dishes. Three days later each dish containing 20 mL medium was treated with vehicle or DHT for 4 h and then was cross-linked with 540 µL 35% formaldehyde for 10 min at room temperature. Glycine was added to dishes, and cells were incubated for 5 min at room temperature. Cells were then washed twice with ice-cold PBS and scraped into PBS + PIC (Protease Inhibitor Cocktail). Pelleted cells were lysed and nuclei were harvested. Nuclear lysates were digested with micrococcal nuclease for 20 min at 37 °C and then sonicated in 500 μL aliquots on ice for 6 pulses of 20 s (with 20 s gaps) using a Branson sonicator. Lysates were clarified at 10,000× *g* for 10 min at 4 °C, and chromatin was stored at −80 °C. Androgen Receptor polyclonal rabbit antibody (Santa Cruz Biotechnology, Dallas, TX, USA; D2C9), EZH2 monoclonal rabbit antibody (Cell Signaling Technology, Danvers, MA, USA; N-20), and IgG antibody (Cell Signalling Technology, Inc, Danvers, MA, USA; 2729), were incubated overnight at 4 °C with rotation and protein G magnetic beads were incubated 2 h at 4 °C. Washes and elution of chromatin were performed following the manufacturer’s instructions. DNA quantification was carried out using a Viia7 Real-Time PCR System (ThermoFisher, Waltham, MA, USA) with SYBR-Green reagents and primers that amplify Androgen Receptor sites (AR sites) present in CERK sequence as well as KLK3 enhancer region [68] as control (detailed in Appendix A).

### 4.8. Cellular Assays

#### 4.8.1. Cell Growth Analysis

15,000 PC3 cells or 30,000 LNCaP cells were plated in triplicate in 12-well dishes. Twenty-four hours later, the cells considered day 0 were fixed in formalin 10%. The same procedure was performed on the days indicated in Figure 4A and Appendix A. Cell growth was measured by staining with crystal violet (0.1% in 20% methanol) for 45 min. The precipitate was solubilized in 10% acetic acid, and the absorbance was measured at 595 nm.

#### 4.8.2. Invasive Growth

40,000 cells were resuspended in 1ml RPMI with 6 % methylcellulose (Sigma-Aldrich St. Louis, MO, USA; M0387). Drops (25 µL) were pipetted on the cover of a 100 mm culture plate. The inverted plates were incubated at 37 °C and 5% CO_2_ for 72 h. Once formed, spheroids were collected, resuspended in collagen I solution (1.7 mg/mL in DMEM) with or without 20 μM C1P, and seeded in a 24-well plate. Day 0 pictures were taken and complete RPMI media was then added on top of the collagen. Spheroids were maintained at 37 °C and 5% CO_2_ for 7 days and pictures were again taken. The area of the spheroids was measured using FiJi software. Relative invasive growth was quantified as area difference on day 7 minus day 0.

#### 4.8.3. Wound Healing

600,000 LNCaP cells were plated in duplicates in 6-well plates, grown to confluency, and the cell monolayer was wounded with a pipette tip. Photomicrographs of each scratch were obtained at the initial time of wound creation and the same location was photographed every 24 h until completion of the study. Fiji software was used to quantify the area of the wound remaining. This number was then converted to a percentage of the scratch area remaining at each time point.

#### 4.8.4. Migration

24-well plates with 8 μm pore size chambers (Corning Costar #3422) were precoated with 30 μL of fibronectin (0.2 μg/μL dH_2_O, Sigma, St. Louis, MO, US; F2006). Cells (50,000 in 100 μL per insert) were resuspended in a medium supplemented with 0.2% fatty acid-free bovine serum albumin (BSA) and seeded in duplicates to the upper part of the chamber of 24-well plates. 300 μL media with 0.2% BSA with or without 20 μM C1P was added to the lower wells. After the corresponding incubation time (8 to 72 h) at 37 °C and 5% CO_2_, chambers were washed and non-migrated cells were removed with a cotton swab. Filters were fixed with formaldehyde (5% in PBS) and stained with 0.1% crystal violet. Cell migration was assessed by counting the number of migrated cells in a Nikon Eclipse 90i microscope equipped with NIS-Elements 3.0 software. Cells were counted in eight randomly selected microscope fields per filter at 20x magnification.

#### 4.8.5. Anchorage-Independent Growth

6 well-plates were coated with a lower layer of 0.6% agar (SeaKem LE agarose, Lonza, Basel, Switzerland) medium mixture (3 mL/well) and stored at 4 °C for at least 30 min to let the agar solidify. 5000 PC3 cells or 15,000 LNCaP cells per well were suspended in a 0.3% low melting agar (Agarose LM, Pronadisa, Conda, Madrid, Spain) medium mixture and 1 mL/well were plated in duplicates. Plates were stored at 4 °C (around 30 min) to allow the solidification of the upper layer and then incubated at 37 °C in a humidified atmosphere of 5% CO_2_ for 3–4 weeks, until colony detection. Colonies were then quantified using Fiji software (ImageJ version 1.53c).

### 4.9. Bioinformatics Analysis and Statistics

Correlations and enrichment bioinformatics analysis were performed with Cancertool [28]. No statistical method was used to predetermine sample size. The experiments were not randomized. The investigators were not blinded to allocation during experiments and outcome assessment. *n* values represent the number of independent experiments performed or the number of individual mice. For each independent in vitro experiment, at least two technical replicates were used and a minimum number of three experiments were performed to ensure adequate statistical power. For in vitro experiments normal distribution was assumed. One sample *t*-test was applied for one component comparisons with control when one of the two groups did not have a variance and paired *t*-test for matched data in each experiment. One-way ANOVA was used for multiple comparisons. For in vivo experiments a non-parametric Mann–Whitney exact test was used. Unless otherwise stated, data analyzed by parametric tests are represented by the mean ± s.e.m. of pooled experiments and median ± interquartile range for experiments analyzed by non-parametric tests. Two-tail statistical analysis was applied for experimental design without predicted result and one-tail for validation or hypothesis-driven experiments. The confidence level used for all the statistical analyses was 95% (alpha value = 0.05). GraphPad Prism 8 software was used for statistical calculations.

## 5. Conclusions

In this study, we demonstrate the potential of computational analysis using publicly available transcriptional datasets to identify novel metabolic targets of AR in prostate cancer. The repressive control of ceramide kinase by this nuclear receptor exemplifies the complex functions of hormone signaling in this disease.

## Figures and Tables

**Figure 1 cancers-13-04307-f001:**
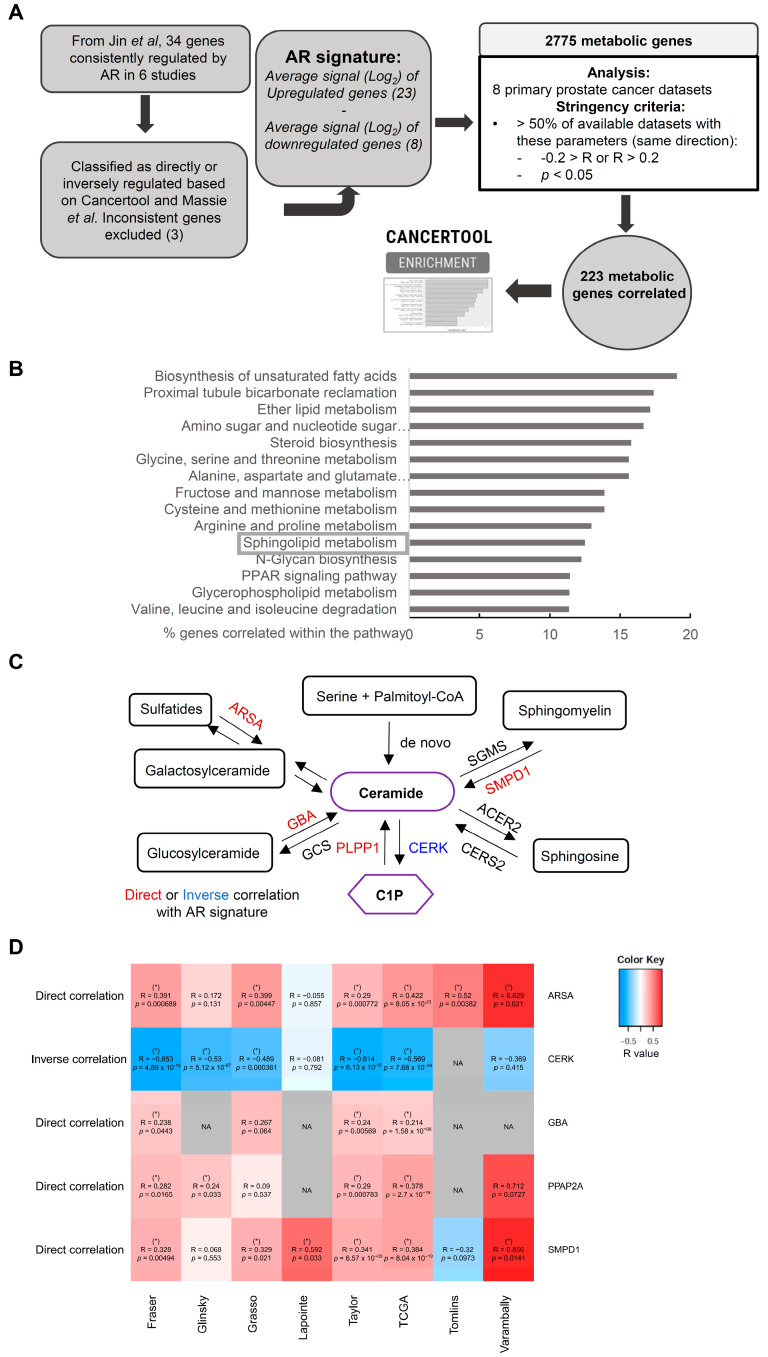
Sphingolipid metabolism is regulated by androgen receptor (AR). (**A**) Bioinformatics analysis workflow of prostate cancer patient transcriptomics datasets performed with *Cancertool* to identify metabolic genes correlated with androgen receptor activity. (**B**) KEGG pathways enrichment of AR activity-correlated metabolic genes (AR metabolic correlome). (**C**) Schematic ceramide biosynthetic pathway, adapted from Merscher et al. Positively and negatively correlated enzymes are highlighted in red and blue, respectively. (**D**) Heatmap representation of the sphingolipid metabolic genes correlated with AR signature in multiple PCa datasets encompassing primary tumors. Red and blue color intensity depict the degree of direct and inverse correlation, respectively. An asterisk is included when the correlation coefficient is greater than 0.2 or lower than −0.2 with a *p*-value lower than 0.05.

**Figure 2 cancers-13-04307-f002:**
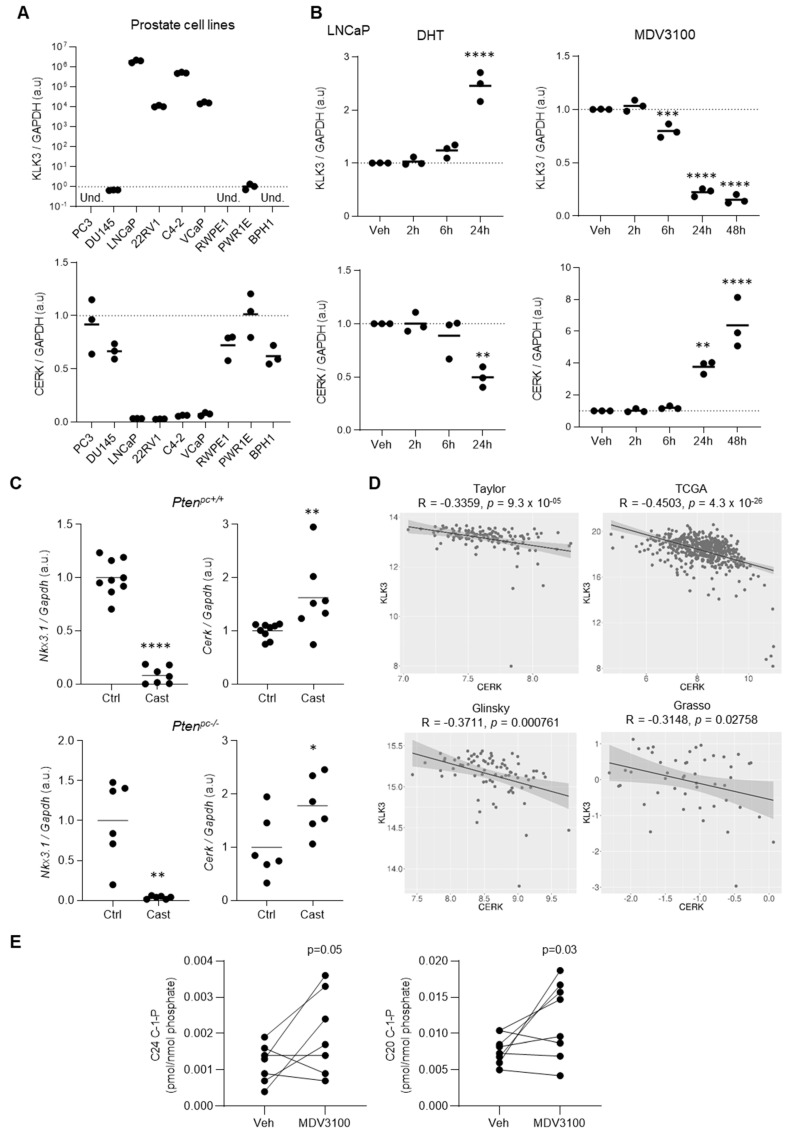
AR represses *CERK* expression. (**A**) Analysis of the AR target *KLK3* and *CERK* gene expression by qRT-PCR in a panel of prostate cells (*n* = 3 independent experiments). Data were normalized to GAPDH expression. mRNA abundance was normalized to the benign cell line PWR1E. Und. stands for undetermined. (**B**) Analysis of *KLK3* and *CERK* gene expression by qRT-PCR upon treatment with AR agonist (dihydrotestosterone, DHT, 10 nM, left panels) or antagonist (MDV-3100, 10 µM, right panels) in LNCaP cells (*n* = 3 independent experiments). Data were normalized to GAPDH expression. ANOVA and Dunnett’s multiple comparisons test was performed. (**C**) Gene expression analysis of the AR target gene, *Nkx3.1*, and *Cerk* by qRT-PCR in prostate-specific *Pten* wild type in upper panels (*Pten^pc+/+^*) castrated at 4 months of age (Cast, *n* = 7 mice) compared to control (Ctl, *n* = 9 mice), and knock out mice in lower panels (*Pten^pc−/−^*, castrated at 6 months of age, *n* = 6 mice per group), 6 days after performing orchiectomy. Data were normalized to *GAPDH/Gapdh* expression. A one-tailed Mann–Whitney test was performed. (**D**) Pearson correlation of *CERK* and *KLK3* mRNA expression in the indicated PCa datasets. (**E**) LC/MS analysis of phosphorylated C24 and C20 ceramide species upon treatment with AR antagonist MDV-3100 (10 µM) compared to vehicle (Veh). Paired t-test analyses were performed. * *p* < 0.05; ** *p* < 0.01; *** *p* < 0.001; **** *p* < 0.0001.

**Figure 3 cancers-13-04307-f003:**
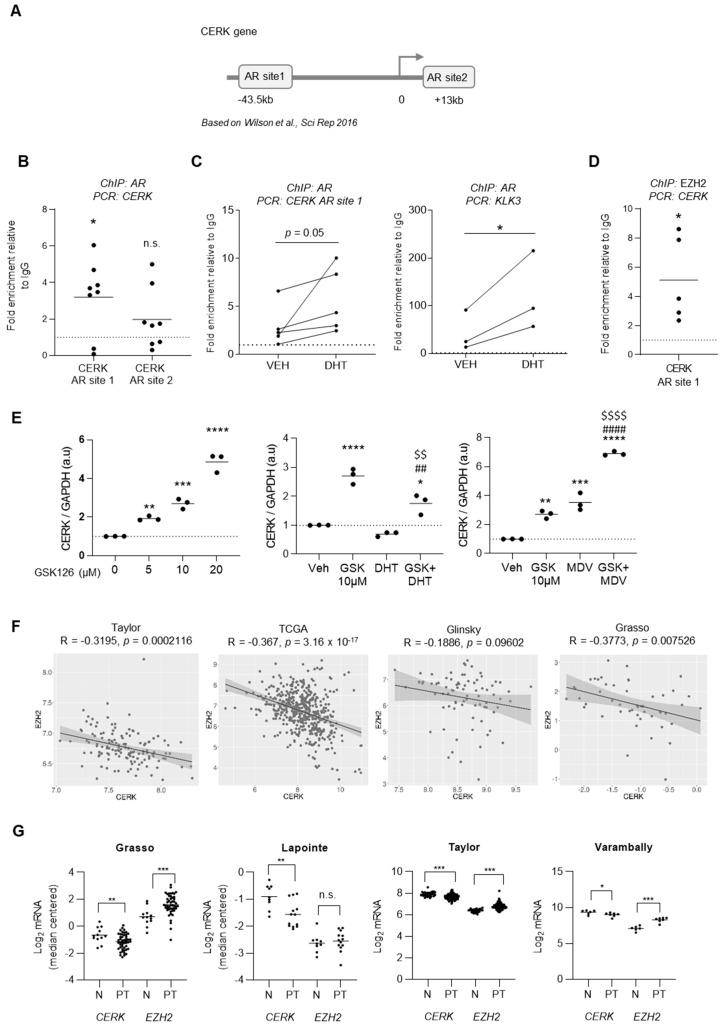
EZH2 cooperates with AR in eliciting CERK repression. (**A**) Androgen receptor binding site (AR sites) location represented on *CERK* gene sequence. (**B**) ChIP of AR on the indicated sites in LNCaP cells (*n* = 8 independent experiments). Data were normalized to IgG (negative-binding control). One-sample *t*-test. (**C**) ChIP of AR on CERK AR site 1 and KLK3 in LNCaP cells (*n* = 3–5 independent experiments) with and without DHT treatment (10 nM). Data were normalized to IgG (negative-binding control). One-tail Paired *t*-test. (**D**) ChIP of EZH2 on AR site 1 of *CERK* regulatory region in LNCaP cells (*n* = 5 independent experiments). Data were normalized to IgG (negative-binding control). One-sample *t*-test. (**E**) Analysis of *CERK* gene expression by qRT-PCR upon 24-h treatment with EZH2 inhibitor GSK126 at increasing concentrations (left), in combination with DHT (10 nM, middle) and MDV-3100 (10 µM, right) in LNCaP cells (*n* = 3 independent experiments). ANOVA with Dunnett´s (left) or Tukey´s (center and right) multiple comparison test. The asterisk refers to comparisons against vehicles, hash refers to comparisons against GSK and dollar refers to comparisons against DHT or MDV. (**F**) Pearson correlation of *CERK* and *EZH2* mRNA expression in the indicated PCa datasets. (**G**) mRNA expression analysis of *CERK* and *EZH2* in normal tissue (N) and localized PCa (PT) of the indicated patient datasets. One tail Student *t*-test. * *p* < 0.05; ** *p* < 0.01; *** *p* < 0.001; **** *p* < 0.0001; ^##^
*p* < 0.01; ^####^
*p* < 0.0001; ^$$^
*p* < 0.01; ^$$$$^
*p* < 0.0001. n.s., non-significant.

**Figure 4 cancers-13-04307-f004:**
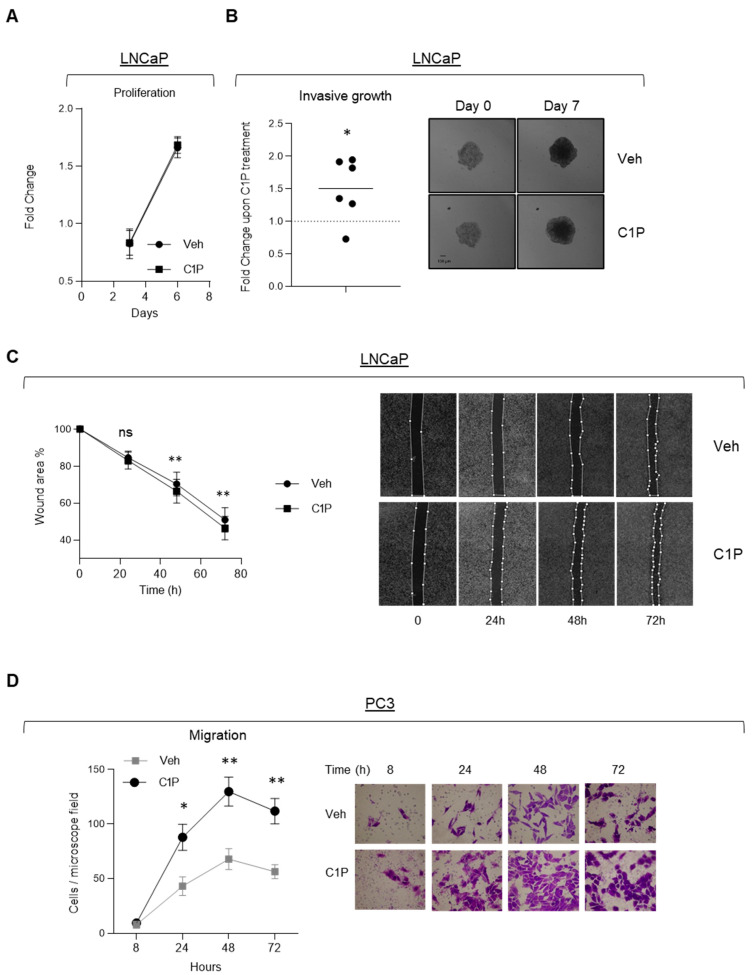
C1P promotes prostate cancer cell migration and invasive growth. (**A**) Proliferation upon treatment with 20 µM ceramide-1-phosphate (C1P) in LNCaP cells (*n* = 3 independent experiments). Paired *t*-test. Comparisons made versus vehicles. (**B**) Quantification of spheroid diameter increase upon treatment with 20 µM C1P in LNCaP cells compared to vehicle (Veh) (*n* = 6 independent experiments), one-sample *t*-test, and representative images of the spheroids at seeding and the end of the experiment. (**C**) Quantification of wound healing migration assay upon treatment with 20 µM C1P in LNCaP cells compared to vehicle (Veh) (*n* = 3 independent experiments, paired *t*-test), and representative images of the wound area at seeding and at the different measured time points. (**D**) Quantification and representative images of cell migration upon treatment with 20 µM C1P in PC3 cells compared to vehicle (Veh) (*n* = 4 independent experiments). Unpaired *t*-test. All comparisons are two-tailed. * *p* < 0.05; ** *p* < 0.01.

## Data Availability

The data presented in this study are available in this article (and Appendix A). The authors are available for any additional inquiries related to the data and the procedures.

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
