# Peer review of "Identification of Androgen Receptor Metabolic Correlome Reveals the Repression of Ceramide Kinase by Androgens"

_cancers, 2021, doi:10.3390/cancers13174307_

Round 1

Reviewer 1 Report

The authors have addressed most of my concerns. However, in order to be accepted for publication, there are still some places that the authors need to address:

  1. It is not clear how the authors identified the AREs located in CERK loci. How did the authors infer the ARE from the referred article.
  2. In figure 3, it is clear that the expression of CERK gene decreases in several patient cohorts, suggesting this gene does not favor the cancer cells. It is not clear why C1P, the product of CERK will increase the cancer invasiveness. Moreover, the phenotypic changes in Figure 4B, C are really hard to appreciate. How was the concentration determined, and which C1P specie was used?
  3. I would suggest putting the relevant LC/MS data in the main figures, as direct evidence of the functional effect of CERK.

Author Response

The authors have addressed most of my concerns. However, in order to be accepted for publication, there are still some places that the authors need to address:

  1. It is not clear how the authors identified the AREs located in CERK loci. How did the authors infer the ARE from the referred article.

We provide the specific citation to the study that reports potential AREs in CERK regulatory region. The AR binding site location and sequences were inferred from the supplementary figure “SI02_CHIPSEQ_ARE_Annotation” from the referred article (Wilson, S.; Qi, J.; Filipp, F.V. Refinement of the androgen response element based on ChIP-Seq in androgen-insensitive and androgen-responsive prostate cancer cell lines. Scientific reports 2016, 6, 32611, doi:10.1038/srep32611.). As suggested by the editor, we renamed these sites as AR sites.

  1. In figure 3, it is clear that the expression of CERK gene decreases in several patient cohorts, suggesting this gene does not favor the cancer cells. It is not clear why C1P, the product of CERK will increase the cancer invasiveness. Moreover, the phenotypic changes in Figure 4B, C are really hard to appreciate. How was the concentration determined, and which C1P specie was used?

We thank the reviewer for this comment. The reviewer is correct, and CERK expression is reduced in prostate cancer tissue. We believe that the explanation is the molecular regulation by androgen receptor reported in this study. We believe that the biological relevance of CERK regulation by AR could be in the context of androgen-deprivation therapy, where CERK de-repression could enhance the survival capacity of cancer cells. We have discussed this aspect in the discussion of the manuscript.

As per the concentration of C1P, we followed the procedures widely reported by us and others (this is the area of expertise of Dr. Antonio Gonzalez-Muñoz, co-author in this study), and we took advantage of C2-Ceramide and C16-Ceramide-1-phosphate. We have clarified this point in the methods section.

  1. I would suggest putting the relevant LC/MS data in the main figures, as direct evidence of the functional effect of CERK.

We thank the reviewer for this suggestion, that we have taken into consideration for the revised version of the manuscript. The relevant data is now included in Figure 2E.

Reviewer 2 Report

The authors answered most of my concerns. However, validation of AR binding in ChIP at the PSA gene (KLK3) is still problematic. This is a key gene targeted by AR, taken as control by the authors. If the authors cannot show nice CHIP and enrichment of AR following androgen stimulation, this raises some doubts about the robustness of all their ChIP. This has to be corrected before publication. In fig. S4E, for their negative control, the treatment induces a 4-5 fold change vs vehicle... a real negative control would not change with DHT treatment. 

Author Response

The authors answered most of my concerns. However, validation of AR binding in ChIP at the PSA gene (KLK3) is still problematic. This is a key gene targeted by AR, taken as control by the authors. If the authors cannot show nice CHIP and enrichment of AR following androgen stimulation, this raises some doubts about the robustness of all their ChIP. This has to be corrected before publication. In fig. S4E, for their negative control, the treatment induces a 4-5 fold change vs vehicle... a real negative control would not change with DHT treatment.

We thank the reviewer for his/her suggestions. To address these points, we have done the following:

  1. We have tested AR binding to KLK3 enhancer with a different set of primers. As presented in the revised manuscript, the increase in binding of AR is significant upon DHT treatment. These data are presented in Fig. 3C.
  2. We have removed AR binding to IDH3A as a negative control. The reviewer is right and there is no information regarding the regulation of IDH3A by AR that ensures that it is a true negative control, and the second potential AR binding site in CERK, that provides negative results in the ChIP assay, has sufficient value as negative control in our view.

Reviewer 3 Report

Most of my concerns have been solved, and the remaining comments do not require experiments, but should be corrected.

1. I requested ChIP data to be presented in a single figure (so when AR ChIP is performed, both negative controls and targets of interest should be presented in the same figure. This enables reader to draw conclusions on how good the specific enrichment is versus authors control ChIP. Now it seems that the specific ChIP is 3-fold enriched and the negative control is 2-fold enriched (comparing CERK ARE1 and CERK ARE2, which the authors nominated as positive and ‘a relevant internal control of the specificity’, respectively). When the data belongs to the same experiment, it must be presented in a single figure.

Please, include ChIP data in the same figure or explain why this can’t be done?

2. ChIP validation is incomplete / data missing / incorrectly presented: Authors state that ‘b. We have included a negative control consisting of gene not regulated by androgens, namely IDH3A. The expression of IDH3A does not respond to AR modulation (Figure

4A for the reviewers´ perusal) and AR does not bind significantly to its promoter (Figure

4B for the reviewers´ perusal). This data is presented in revised Supplementary Figure

4C-E.’

However, Supplementary Figure 4C-E talks about IDH3A and KLK3. Again, just present the ChIP data in the same figure (main or supplementary). If you want to include RT-qPCR data on IDH3A, that can be in supplementary.

3. Authors still refer to an article that is 5 years old as ‘recent’. Find an article that you refer to that is from 2020 at least, so it is relatively recent. Easier even, just don’t call an article recent: articles you select as you references should be significant enough even if they were not published recently.

Author Response

  1. I requested ChIP data to be presented in a single figure (so when AR ChIP is performed, both negative controls and targets of interest should be presented in the same figure. This enables reader to draw conclusions on how good the specific enrichment is versus authors control ChIP. Now it seems that the specific ChIP is 3-fold enriched and the negative control is 2-fold enriched (comparing CERK ARE1 and CERK ARE2, which the authors nominated as positive and ‘a relevant internal control of the specificity’, respectively). When the data belongs to the same experiment, it must be presented in a single figure. Please, include ChIP data in the same figure or explain why this can’t be done?

We would like to apologize for missing this point in the previous revision. As requested, we have included both potential AR binding sites in CERK in Fig. 3B and all ChIP data in the same main figure (Fig. 3).

  1. ChIP validation is incomplete / data missing / incorrectly presented: Authors state that ‘b. We have included a negative control consisting of gene not regulated by androgens, namely IDH3A. The expression of IDH3A does not respond to AR modulation (Figure 4A for the reviewers´ perusal) and AR does not bind significantly to its promoter (Figure4B for the reviewers´ perusal). This data is presented in revised Supplementary Figure4C-E.’ However, Supplementary Figure 4C-E talks about IDH3A and KLK3. Again, just present the ChIP data in the same figure (main or supplementary). If you want to include RT-qPCR data on IDH3A, that can be in supplementary.

We thank the reviewer for this clarification. Based on the comments of another reviewer, we have considered that we do not have sufficient data to refer to IDH3A as a negative control, and therefore we have chosen to remove this data.

  1. Authors still refer to an article that is 5 years old as ‘recent’. Find an article that you refer to that is from 2020 at least, so it is relatively recent. Easier even, just don’t call an article recent: articles you select as you references should be significant enough even if they were not published recently.

The reviewer is right, and we unintentionally left this phrase in the text. We have removed the term recent.

Round 2

Reviewer 2 Report

The authors corrected the final concerns I had.

This manuscript is a resubmission of an earlier submission. The following is a list of the peer review reports and author responses from that submission.

Round 1

Reviewer 1 Report

This paper identifies a gene, CERK, that is regulated by androgen signaling. There is limited novelty, but overall figures and data are presented clearly and language of the paper is good. I have made a number of comments, most of which are easy to correct (always, make the correction in the article you are trying to get published, or explain why you think my proposed change is not good). The bioinformatics approach doesn’t bring much new – all these are already known, as authors acknowledge. Authors select one process that hasn’t been previously studied too much, because it has not been top-enriched in previous studies either. I propose couple of experiments that are essential to get my approval for this paper.

Results

  1. Experiments using siRNA (or shRNA or CRISPR) knockdown of the CERK enzyme in androgen receptor-dependent prostate cancer cells is needed (at least two prostate cancer cell line models). The data produced in the cancer cell lines has to be compared to normal cells. Otherwise, this is a study that identifies androgen receptor-regulated mRNA, which without functional validation has limited remit, and in my mind is not sufficient for publication.

  1. Confirm that CERK is androgen regulated in the protein level (western blot or similar).

  1. Androgen-dependent / anti-androgen-dependent regulation of CERK is only evaluated in one cell line (LNCaP and C4-2; C4-2 being a derivative of LNCaP; please use VCaP or 22RV1 or LAPC4 to confirm your results in a cell line that is not a derivative of LNCaP).

  1. Authors should include a negative site for AR ChIPs (look for example from these papers: PMID: 28591577 or PMID: 28412251)

  1. It is not at all clear why EZH2 is introduced in the study. Consider removing these data or adding new data: Please, explain why EZH2 inhibitor has selective effects on CERK expression but not on KLK3 expression. I propose authors do here re-ChIP experiments (first ChIP for AR and followed by ChIP for EZH2 and qPCR for KLK3 AR binding site and CERK binding site. This way authors can have a mechanism why AR/EZH2 axis differentially regulates CERK/KLK3.

  1. When comparing KLK3 and CERK expression and ChIPs, please include all the data in the main figures.

  1. Authors jump from AR signature to a group of metabolic genes without any logic (try at least start a new chapter to make a clear-cut transition.

  1. Cancertool lacks references.

  1. What is invasive growth of organoids (explanation of these data should be in the results section)?

  1. How much of sphingoine-1-phosphate was added (how was this determined and is this dose physiologically relevant)? Please, include info in the figure or main text. Also, include example images of your spheroids.

  1. Authors should use sphingosine as a negative control in these experiments to show that it is the phosphorylated form of sphingosine that is important.

  1. Why do authors use androgen receptor negative PC3 cells in their migration experiments with sphingosine-1-phosphate, even though the whole premise of the paper was to figure out androgen receptor-dependent effects?

  1. Figure 3A. What is ‘one-sample t test’? Why don’t you use normal t-test (paired sample seems appropriate here, and certainly 2-tailed).

  1. Figure 3B. Is t-test 2-tailed? Explain in the figure legend. Also, if not 2-tailed, explain to reviewer in your response letter.

  1. Authors should describe the AR-dependent regulation (Figures 1, 2 and 4 first), and then progress to data presented in Figure 3.

  1. Authors state ‘We asked whether CERK expression could be directly repressed by AR.’, and used ChIP. ChIP only demonstrates if the factor of interest (AR here) binds to site of interest. It will not explain regulatory relationship.

  1. Can authors explain / re-phrase ‘submaximal’? What does this word mean in the scientific context?

Abstract

  1. In abstract, it sounds like authors claim that AR binds to CERK (even though they most likely refer to genomic element in the CERK promoter where AR binds and thereby regulates CERK expression).

  1. It is unclear why authors bring up EZH2?

  1. Biochemistry as a key word is very broad.

Introduction

  1. What do authors mean by statement ‘In male reproductive organs, androgens activate androgen receptor (AR) to elicit discreet transcriptional programs [4]’? Discreet typically refers to a person rather than a transcription factor.

  1. What are ‘these lipidic signals’ that pop-up out of nowhere in the introduction?

  1. What are processes that are ‘negative for cell function’?

Discussion

  1. Please avoid words ‘recent’, as they don’t stand time very well. Especially when referring to a study that was published 2013, you can’t call it recent.

Methods

  1. Explain these abbreviations: CIC, bioGUNE, AAALAC

  1. Authors state that ‘Mice were fasted for 6 h prior to tissue harvest in order to prevent metabolic alterations due to immediate food intake.’. Why is this important? Did authors perform some metabolic analysis on these samples? Please, include in you paper these data.

  1. When you use commercially available kits, there’s no need to refer to a paper on how to use it unless you deviate from the protocol manufacturer gives you. If you do deviate, please explain that deviation in the method-section of this paper rather than making the reader to seek the other paper.

  1. Crosslinking ChIP with 35% paraformaldehyde for 10 minutes is pretty rough. About 35 times higher concentration than what is typically used for the same time. Is this correct?

Supplementary

  1. Please have the supplementary figure legends underneath each figure to make it easier to see what is done.

  1. Explain all the abbreviations when first used.

  1. Explain in the legend for Supplementary table 3 how this table was generated with appropriate references.

Author Response

Reply enclosed

Reviewer 2 Report

In this study “Identification of androgen receptor metabolic correlome reveals the repression of ceramide kinase by androgens”, Camacho et al., identified by computational analysis of publicly available transcriptional datasets, the association between AR and CRK expression. I particular they show that AR has a repressive action on CERK mRNA. In addition, by using cell and mouse models, they show that such a repressive effect of AR occurs also in vivo in strict association with EZH2 activity. The study is interesting, however a major concern raises reading this manuscript.

Indeed, previous works have shown that C1P exerts a mitogenic effect by activating several signaling pathways such as ERK, PI3K, JNK and KFkB. Moreover is able to induce stimulated macrophage proliferation. In view of these findings, repression of CERK, as suggested by the Authors, causes a reduction of C1P, that is counterintuitive as respect to its positive effect on cancer growth. Therefore, meanwhile the Authors suggested that the “pathophysiological context would this regulation be relevant remains to be defined”, I think that the main paper observation, the negative regulation of CERK, must be associated with the measure of C1P levels as well as ceramide and S1P levels. These data cuold strongly help to find a rationale in such a negative regulation in prostate cancer.

2) The migration experiments have to be associated to a proliferation analysis, in order to exclude that the enhanced migration is associated with a major cell number.

3) I suggest also to make a better discussion especially adding some of the numerous references in which the role of EZH2 in AR repressive effect has been previously described, see for instance

- Zhao JC, Yu J, Runkle C, Wu L, Hu M, et al. Cooperation between Polycomb and androgen receptor during oncogenic transformation. Genome Res.

- CCN3/NOV gene expression in human prostate cancer is directly suppressed by the androgen receptor. Oncogene. 2014

- Fong KW, Zhao JC, Kim J, Li S, Yang YA, et al. Polycomb-mediated disruption of an androgen receptor feedback loop drives castration-resistant prostate cancer. Cancer Res. 2017;77:412–22.

- Chng KR, Chang CW, Tan SK, Yang C, Hong SZ, et al. A transcriptional repressor co-regulatory network governing androgen response in prostate cancers.

Author Response

Reply enclosed

Reviewer 3 Report

The study by Camacho et al. describes a new metabolic pathways regulated by AR in prostate cancer. It is an exciting new area of research rapidly growing and the study is clearly of interest in Cancers. However, before being suitable for publication, a few major comments need to be addressed.

1. The authors clearly demonstrate that CERK is an androgen-sensitive gene. Yet, the impact of the study would be much higher if they could demonstrate that AR modulation significantly alters CERK activity (clearly linking mRNA regulation to functional modulation).

2. The authors show ChIP data of AR binding at the CERK gene. Fold enrichment are actually pretty low; yet, it does not mean that they are not biologically relevant. However, additional controls are required, such as performing AR ChIP with and without androgen (the later being expected to be lower). Additionnal ChIP of EZH2 at CERK, with and without androgen, could also be performed to show recruitment of EZH2 at this locus by AR.

3. The discussion is rather short. I would have liked to have more details regarding PCa literature, notably given that it is well known now that AR induces lipid synthesis AND fatty acid beta-oxidation, an apparent paradox that still has not been elucidated. Yet, there is plenty of literature on that matter that could have been discussed to better put in perspective their work. For example, the authors should refer to articles showing that the lipid regulatory transcription factor, SREBF1, is also regulated by AR, which modulates indirectly lipid metabolism in prostate cancer cells. 

Author Response

Reply enclosed

Reviewer 4 Report

In this study, the authors aim to investigate the Androgen Receptor (AR) activity correlation with the metabolic program in prostate cancer. They first took a computational approach to build a AR gene signature, and then performed correlation analysis of this signature and well curated metabolic genes. Among the metabolic pathways, They were interested in the sphingolipid metabolism, which regulates the turnover of ceramide. The enzyme CERK that phosphorylates ceramide and promotes cell invasion was negatively correlated with AR activity. The authors then started to study the AR direct regulation on CERK, validated this negative correlation in cell lines and also in the public database. Using GSK126 inhibitor, it was proposed that EZH2 plays a role in the AR repressive regulation of CERK. Overall, the study is interesting and provides some clear evidence of a maintained inverse relationship between AR and CERK. However, it also misses some key experiments to support the conclusions.

1) The whole study only measured the CERK mRNA level. The protein level of CERK should be demonstrated at least in some experiments. 

2) To demonstrate the repressive function of AR on CERK, VCaP cell line is the most appropriate model. In fact, C4-2 is better than LNCap, as CERK expression responded to DHT in a quicker manner. Moreover, the weak AR binding signal in the ChIP experiment shows that AR might not directly repress CERK in LNCaP. Also, it is not clear what is the condition for the ChIP, was there DHT treatment, how long? There are many public AR ChIP-seq in different prostate cancer cell lines. The authors can strengthen the evidence of AR chromatin binding by exploiting these datasets. 

3)To conclude that "EZH2 cooperates with AR", the authors should at least provide evidence of EZH2 binding on the ARE together with AR. Otherwise, a more broad conclusion should be drawn.

4) The correlation analysis in Fig2D is a manifestation of Fig1D, since KLK3 is one of the AR signature genes. Instead, the AR vs CERK should be looked at.

5) The functional study in Fig3 should also include CERK inhibitor and DHT treatment. It is not clear what is the rationale of using AR-negative PC3 cells, which express high level of CERK.

6) There is a lack of discussion on how AR suppression of CERK could contribute to PCa progression to CRPC where AR activity has been blocked and reactivated. Line 252 "...beyond the context of AR", how this is concluded? It is inconsistent with the statement suggesting EZH2 cooperates with AR. 

7) There should be more detail about the cell culture condition in the experiments, were LNCap in hormone-depleted medium, and how long.

Author Response

Reply enclosed
